# The compounding effect of having HIV and a disability on child mortality among mothers in South Africa

**Ilhom Akobirshoev**[1]*, **Hussaini Zandam**[1], **Allyala Nandakumar**[1], **Nora Groce**[2], **Mark Blecher**[3], **Monika Mitra**[1]

**1** The Lurie Institute for Disability Policy, The Heller School for Social Policy and Management, Brandeis University, Waltham, Massachusetts, United States of America, **2** UCL International Disability Research Centre, University College London, London, United Kingdom, **3** National Treasury, Pretoria, South Africa

* ilhom@brandeis.edu

## Abstract

### Background

Previous research on the association between maternal HIV status and child mortality in sub–Saharan Africa was published between 2005–2011. Findings from these studies showed a higher child mortality risk among children born to HIV–positive mothers. While the population of women with disabilities is growing in developing countries, we found no research that examined the association between maternal disability in HIV–positive mothers, and child mortality in sub–Saharan Africa. This study examined the potential compounding effect of maternal disability and HIV status on child mortality in South Africa.

### Methods

We analyzed data for women age 15–49 years from South Africa, using the nationally representative 2016 South Africa Demographic and Health Survey. We estimated unadjusted and adjusted risk ratios of child mortality indicators by maternal disability and maternal HIV using modified Poisson regressions.

### Results

Children born to disabled mothers compared to their peers born to non-disabled mothers were at a higher risk for neonatal mortality (RR = 1.80, 95% CI:1.31–2.49), infant mortality (RR = 1.69, 95% CI:1.19–2.41), and under-five mortality (RR = 1.78, 95% CI:1.05–3.01). The joint risk of maternal disability and HIV-positive status on the selected child mortality indicators is compounded such that it is more than the sum of the risks from maternal disability or maternal HIV-positive status alone (RR = 3.97 vs. joint RR = 3.67 for neonatal mortality; RR = 3.57 vs. joint RR = 3.25 for infant mortality; RR = 6.44 vs. joint RR = 3.75 for under-five mortality).

### Conclusions

The findings suggest that children born to HIV-positive women with disabilities are at an exceptionally high risk of premature mortality. Established inequalities faced by women with

**Data Availability Statement:** All 2016 Demographic & Health Survey data used for this study are publicly available can be accessed from the https://dhsprogram.com/methodology/survey/

survey-display-390.cfm?showall=yes after registration.

**Funding:** Funding support for this research was provided by Cardno Emerging Markets, USA, Ltd, as part of the Public–Private Partnerships in PEPFAR Countries project of the Centers for Disease Control and Prevention (CDC), Division of Global HIV/AIDS and TB (DGHT) under Cooperative Agreement Number U2GGH001531. The funder did not participate in the study design; in the collection, analysis, interpretation of data; or in the writing and submission of the manuscript for publication.

**Competing interests:** I have read the journal's policy and the authors of this manuscript have the following competing interests: Drs. Akobirshoev, Zandam and Nandakumar report receiving a grant from Cardno Emerging Markets, USA, Ltd., for the conduct of the study. I hereby confirm that this does not alter our adherence to PLOS ONE policies on sharing data and materials.

disabilities may account for this increased risk. Given that maternal HIV and disability amplify each other's impact on child mortality, addressing disabled women's HIV-related needs and understanding the pathways and mechanisms contributing to these disparities is crucial.

## Introduction

About 15 percent of the world's population, or over 1 billion people, live with disabilities, and 80 percent live in low- and middle-income countries (LMICs) [1]. Growing evidence points to a higher risk of human immunodeficiency virus (HIV) among people with disabilities living in the Global South. Prior research [2], including a recent study by De Beaudrap and colleagues [3], have found consistent and strong associations between disability status and exposure to HIV.

Several studies have also pointed to the greater risk of HIV infection among women with disabilities in several sub-Saharan African countries [4]. Girls and women with disabilities are among the most socially and economically marginalized populations in the Global South [3]. They have less access to educational and employment opportunities, experience difficulty with accessing health care services, and are more vulnerable to gender-based violence in their homes and communities [2, 5–8]. Disabled women face discriminatory attitudes regarding their sexuality and ability to parent, and are often neglected in the planning and implementation of sexual and reproductive health program programs including in HIV prevention, diagnosis, and treatment programs [3, 9]. In their Lancet commentary, Tun and Leclerc-Madlala [10] posit that despite including HIV and disability in national strategic plans in Africa, one of the reasons for inaction by governments, donors, and non-governmental organizations (NGOs) to invest in disability-inclusive HIV programming is the lack of robust epidemiological studies on the intersection of HIV and disability.

An overwhelming body of evidence suggests higher rates of child mortality among children born to women with HIV [11, 12]. Despite the increased vulnerability to HIV infections in disabled women, there are no epidemiological studies investigating child mortality rates among HIV-positive women with disabilities. Children born to women with disabilities who are HIV-positive likely experience "double jeopardy," resulting in an even higher risk of child mortality than risks associated with maternal disability or maternal HIV alone.

The purpose of this study is to address this paucity of epidemiology studies on the intersection of maternal disability and HIV by examining the potential compounding effect of maternal disability status and maternal HIV status on the risk of child mortality in South Africa. The early adoption by South Africa Demographic and Health Surveys (SADHS) of the Washington Group Disability Questions and collection of biomarkers for HIV testing, allowed us to examine these compounded risks among women in South Africa. South African is considered one of Africa's leading countries in their championing of disability in their national strategic plans and in their early adoption of disability-inclusive HIV programs and services [13]. Finding compounding risk in disability-inclusive service ecosystems of South Africa could emphasize the importance of prioritizing concentrated public health efforts and HIV prevention and treatment services towards women with disabilities who are HIV-positive and their children who experience a double burden of child mortality risks.

We hypothesized that the combined effect of maternal disability status and maternal HIV status on select child mortality indicators will be compounded, i.e., greater than either maternal disability status and maternal HIV status alone.

## Methods and methods

### Data

We used cross-sectional data from the 2016 South Africa Demographic and Health Surveys (SADHS) [14]. The SADHS provides up-to-date estimates of key demographic, socioeconomic, and health indicators in South Africa, including sexual and reproductive health in adults, infant and maternal mortality, child mortality, nutritional status, malaria, disability status, and biomarkers including HIV status. Detailed information about survey design, sampling methods, and response rates are available in the SADHS final survey reports [14].

### Sample

The SADHS data are nationally representative of South African women 15–49 years of age. A total of 8,514 women were interviewed in 2016 (see Fig 1). Of these, only 2,726 were eligible for HIV testing and were both interviewed and tested. We excluded women who have never given birth to a living child (N = 719) or women who had missing or inconclusive HIV test results (N = 23). Our final analytic sample included 1,984 mothers, aged 15–49 and their 4,667 liveborn children.

### Outcomes

The outcome variables included child mortality indicators, including (1) neonatal mortality, (2) infant mortality, and (3) under-five mortality. The selected child mortality indicators were measured using the information from SADHS birth history on whether the liveborn child was alive or dead at the time of interview and age at death (if died). Neonatal, infant and under-five mortality indicators were measured based on mothers' reports during the interview on whether any of their liveborn children died before reaching one month, 12 months, and 60 months of age, respectively. These indicators excluded cases of stillbirths, miscarriages, or abortions. All outcome variables were measured as binary variables (i.e., yes or no).

### Exposure

Having a disability and having HIV were considered as risk factors. Disability status is measured as a binary indicator (i.e., yes or no). We categorized women as having a disability if they reported "a lot of difficulty" or "cannot function at all" to any of the Washington Group Short Set of Questions on Disability [15] functional areas related to 1) seeing, 2) hearing, 3) communicating, 4) remembering, 5) walking, and 6) washing or dressing. Exposure to HIV is measured as a binary indicator (i.e., positive or negative) indicating HIV infection. Dried blood spot samples were collected from randomly selected women in the household and tested for HIV. We combined disability and HIV indicators and created a new variable with the following categories: (1) women with disabilities who are HIV-negative, (2) women without disabilities who are HIV-positive, and (3) women with disabilities who are HIV-positive.

### Covariates

We included the following sociodemographic characteristics as covariates in all our multivariate analyses: age (< 25 years, 25–34 years, 35+ years), education (no education, primary, secondary, higher), marital status (never married, formerly married/widowed, married/partnered), number of currently living children (0, 1, 2, 3, 4, or more), and employment status (i.e., employed or unemployed). Household characteristics included household wealth quintile (lowest, second, third, fourth, highest), and residence (i.e., urban or rural). Sex of the live-born child was also included as a covariate.

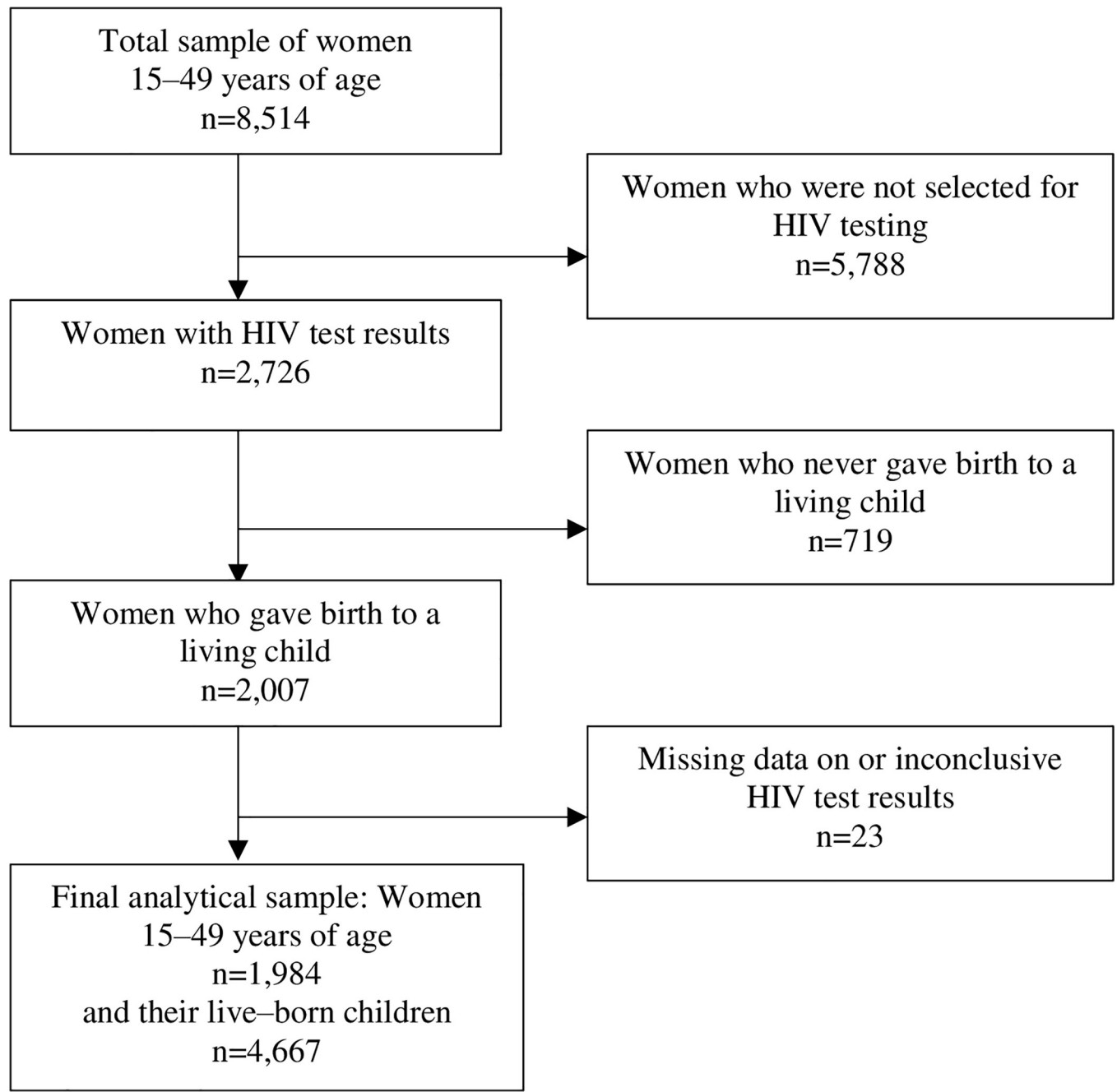

**Fig 1. Analytic sample selection, South Africa Demographic Health Survey (SADHS) 2016.** This flow chart depicts the steps in narrowing the sample from the full number of women 15–49 years of age available in the data source to the number included in our analyses. South Africa Demographic Health Survey (SADHS) 2016 [14].

### Statistical analysis

We compared demographics and socioeconomic characteristics of women with and without disabilities by HIV status. Differences between mothers with and without disabilities who gave birth to a live-born child were evaluated using the chi-square test for categorical and the t-test for continuous variables. The sample of women who ever gave birth to a live-born child (N = 1,984) was used for descriptive analysis.

All child mortality indicators were analyzed as binary variables (i.e., yes or no), coded as higher prevalence indicating a greater risk of child mortality. We calculated the prevalence for each child mortality indicators by maternal disability and HIV status and estimated unadjusted and adjusted risk ratios using modified Poisson regressions. Women without disabilities who were HIV-negative, with neither of these risk factors served as the referent group. We analyzed cohorts of (1) mothers with disabilities who were HIV-negative, (2) mothers without disabilities who were HIV-positive, and (3) mothers with disabilities who were HIV-positive. The sample of live-born children (N = 4,667) was used in all regression analyses. Multivariable models adjusted for the covariates described above. Appropriate adjustment for individual-level clustering due to the inclusion of live-born children to the same mother was made using the robust clustered sandwich estimator method [16]. We used Stata version 16 (StataCorp LLC, College Station, Texas, USA) for all analyses, applying svy commands to account for the SADHS's complex sampling design.

## Results

Table 1 presents the demographic and socioeconomic characteristics of mothers with and without disabilities who gave birth to a live-born child, stratified by HIV status. In each HIV and non-HIV cohort, mothers with disabilities were more likely to be older and less likely to be employed than their counterparts without disabilities. We found no other significant differences between mothers with and without disabilities, both in HIV and non-HIV cohorts, for other characteristics.

Table 2 reports rates, unadjusted and adjusted risk ratios, and respective 95% confidence intervals for child mortality indicators by maternal disability and HIV status.

### Maternal disability status and child mortality

Among HIV-negative mothers the rates of the three selected indicators of child mortality were nearly 2-fold higher among disabled mothers compared to their non-disabled counterparts (73.7 vs. 40.9 per 1,000 live births for neonatal mortality, 59.9 vs. 35.4 per 1,000 live births for infant mortality, and 28.9 vs. 16.3 per 1,000 live births for under-five mortality). In unadjusted analyses, HIV-negative mothers with disabilities were at a higher risk for neonatal mortality (RR = 1.80, 95% CI: 1.31–2.49), infant mortality (RR = 1.69, 95% CI: 1.19–2.41), and under-five mortality (RR = 1.78, 95% CI: 1.05–3.01). The risks remained robust and statistically significant even after adjustment for maternal sociodemographic characteristics and sex of the child.

### Maternal HIV status and child mortality

Although we found nearly two-fold higher rates of child mortality indicators in mothers without disabilities who were HIV-positive than in mothers without disabilities who were HIV-negative (76.4 vs. 40.9 per 1,000 live births for neonatal mortality, 55.3 vs. 35.4 per 1,000 live births for infant mortality, and 32.1 vs. 16.3 per 1,000 live births for under-five mortality), results from our unadjusted and adjusted multivariate analyses show that these differences did not reach statistically significant levels.

### The compounded effect of maternal disability and maternal HIV on child mortality

Compared to child mortality indicators in non-disabled mothers who were HIV-negative, the rates were four-fold higher for neonatal mortality (162.3 vs. 40.9 per 1,000 live births), 3.6-fold higher for infant mortality (126.4 vs. 35.4 per 1,000 live births), and 6.4-fold higher for under-five mortality (105.1 vs. 16.3 per 1,000 live births) in mothers with disabilities who were HIV-

**Table 1. Description of the sample of mothers 15–49 years old who gave birth to a live-born child by disability and HIV status, South Africa, 2016, N = 1,984 (weighted percentages).**

| Characteristics | HIV-negative n = 1,392 Disability status | | p-value | HIV-positive n = 645 Disability status | | p-value |
|---|---|---|---|---|---|---|
| | No n = 1,297 | Yes n = 42 | | No n = 613 | Yes n = 32 | |
| **Maternal characteristics** | | | | | | |
| Age at birth | | | **0.008** | | | 0.705 |
| <25 | 21.8 | 7.0 | | 9.5 | 5.7 | |
| 25–34 | 37.8 | 21.4 | | 44.2 | 40.9 | |
| 35+ | 40.4 | 71.5 | | 46.3 | 53.4 | |
| Age, mean (SE) | 32.5(0.3) | 38.5(1.4) | **0.01** | 34.0(0.4) | 35.2(1.7) | 0.723 |
| Highest educational level | | | 0.280 | | | 0.624 |
| No education | 1.9 | 6.1 | | 3.0 | 2.3 | |
| Primary | 10.6 | 10.9 | | 13.0 | 13.4 | |
| Secondary | 75.8 | 74.3 | | 78.3 | 84.2 | |
| Higher | 11.7 | 8.7 | | 5.7 | 0.0 | |
| Current marital status | | | 0.738 | | | 0.774 |
| Never married | 49.4 | 43.3 | | 57.5 | 58.2 | |
| Married | 44.0 | 50.2 | | 35.3 | 31.6 | |
| Formerly married | 6.6 | 6.4 | | 7.2 | 10.2 | |
| Number of currently living children | | | 0.073 | | | 0.672 |
| 0 | 1.1 | 0.0 | | 2.5 | 3.5 | |
| 1 | 36.9 | 28.0 | | 35.3 | 32.0 | |
| 2 | 31.2 | 15.6 | | 31.6 | 22.5 | |
| 3 | 16.9 | 34.3 | | 20.5 | 30.0 | |
| 4 | 8.6 | 19.6 | | 4.8 | 2.3 | |
| 5+ | 5.2 | 2.6 | | 5.3 | 9.6 | |
| Currently working | | | **0.040** | | | **0.023** |
| No | 62.9 | 42.2 | | 61.7 | 81.9 | |
| Yes | 37.1 | 57.8 | | 38.3 | 18.1 | |
| **Household characteristics** | | | | | | |
| Wealth index | | | 0.719 | | | 0.922 |
| Poorest | 21.1 | 18.4 | | 24.0 | 31.4 | |
| Poorer | 19.2 | 10.8 | | 22.2 | 15.6 | |
| Middle | 23.2 | 21.6 | | 24.9 | 24.4 | |
| Richer | 19.1 | 25.3 | | 19.7 | 19.2 | |
| Richest | 17.4 | 23.9 | | 9.3 | 9.4 | |
| Place of residence | | | 0.747 | | | 0.964 |
| Urban | 67.4 | 64.4 | | 67.1 | 67.5 | |
| Rural | 32.6 | 35.6 | | 32.9 | 32.5 | |
| **Child characteristics** | | | | | | |
| **No. of liveborn children, N = 4,667** | | | | | | |
| No. of liveborn children, Mean (SE) | 2.9(0.1) | 3.2(0.2) | 0.228 | 2.8(0.1) | 3.1(0.4) | 0.445 |
| Sex of the child | | | 0.799 | | | 0.356 |
| Male | 51.7 | 54.8 | | 55.4 | 44.7 | |
| Female | 48.3 | 45.2 | | 44.6 | 55.3 | |

Source: South Africa Demographic and Health Survey (SADHS) 2016 [14]. P-values for differences are based on Chi2-test, or t-test. Boldface indicates statistical significance (p<0.05). HIV denotes the human immunodeficiency virus; No. denotes number; SE denotes a standard error.

**Table 2. Rates, unadjusted and adjusted risk ratios (with 95% confidence intervals) for child mortality indicators by maternal disability and HIV status, N = 4,667 live-born children to 1,984 mothers.**

| Child mortality indicator | No Disability | | Disability | | No Disability | | Disability | |
|---|---|---|---|---|---|---|---|---|
| | HIV-negative | | HIV-negative | | HIV-positive | | HIV-positive | |
| Number of women | 1,297 | | 42 | | 613 | | 32 | |
| Number of live born children | 2,964 | | 123 | | 1,491 | | 89 | |
| **Neonatal mortality (died before 1 mo.)** | | | | | | | | |
| Per 1,000 live birth [a] (95% CI) | 40.9 | (31.1–53.7) | 73.7 | (56.1–96.4) | 76.4 | (34.1–162.4) | 162.3 | (73.9–320.2) |
| uRR, (95% CI) | Referent | | **1.80**[***] | **(1.31–2.49)** | 1.87[*] | (0.96–3.64) | **3.97**[***] | **(2.16–7.27)** |
| aRR[b], (95% CI) | Referent | | **1.73**[***] | **(1.25–2.39)** | 1.90[*] | (0.98–3.68) | **3.35**[***] | **(1.91–5.87)** |
| **Infant mortality (died before 12 mo.)** | | | | | | | | |
| Per 1,000 live birth, (95% CI) | 35.4 | (26.5–47.1) | 59.9 | (43.5–81.9) | 55.3 | (20.1–142.9) | 126.4 | (44.6–309.6) |
| uRR, (95% CI) | Referent | | **1.69**[***] | **(1.19–2.41)** | 1.56 | (0.71–3.45) | **3.57**[***] | **(1.75–7.28)** |
| aRR, (95% CI) | Referent | | **1.65**[***] | **(1.15–2.35)** | 1.63 | (0.74–3.56) | **3.03**[***] | **(1.58–5.84)** |
| **Under-five mortality (died before 60 mo.)** | | | | | | | | |
| Per 1,000 live birth, (95% CI) | 16.3 | (10.5–25.2) | 28.9 | (16.4–50.7) | 32.1 | (10.7–92.0) | 105.1 | (39.6–250.3) |
| uRR, (95% CI) | Referent | | **1.78**[**] | **(1.05–3.01)** | 1.97 | (0.80–4.82) | **6.44**[***] | **(2.89–14.36)** |
| aRR, (95% CI) | Referent | | **1.73**[**] | **(1.02–2.93)** | 2.15 | (0.85–5.43) | **5.38**[***] | **2.55–11.36)** |

Source: South Africa Demographic and Health Survey (SADHS) 2016 [14].

[***] p < 0.01

[**] p < 0.05

[*] p < 0.1. Notes

[a]Weighted rates

[b]Adjusted for maternal age, education, marital status, number of living children, employment, household wealth, residence, and sex of the liveborn child. Boldface indicates statistical significance (p<0.05). Abbreviations: uRR = unadjusted risk ratios, aRR = adjusted risk ratios, CI = confidence interval, mo = months, HIV = human immunodeficiency virus.

positive. Unadjusted analyses showed that compared to mothers without disabilities who were HIV-negative, mothers with disabilities who were HIV-positive had nearly four times higher risk for having their children die during the first month (RR = 3.97, 95% CI: 2.16–7.27), 3.6 times during the first year (RR = 3.57, 95% CI: 1.75–7.28), and 6.4 times during the five years of their life (RR = 6.44, 95% CI: 2.89–14.36). The risks were only slightly attenuated and remained robust and statistically significant, even after adjusting for maternal sociodemographic characteristics and sex of the child.

These findings indicate that the joint risk of maternal disability and maternal HIV-positive status on the selected child mortality indicators is compounded such that it is more than the sum of the risks from maternal disability or maternal HIV-positive status alone (RR = 3.97 vs. joint RR = 3.67 for neonatal mortality, RR = 3.57 vs. joint RR = 3.25 for infant mortality, and RR = 6.44 vs. joint RR = 3.75 for under-five mortality).

The rates of child mortality indicators in disabled mothers and in mothers with HIV were nearly twice the rates as their non-disabled counterparts. As we expected, the rates were even higher in disabled mothers with HIV, including almost four-fold higher for neonatal and infant mortality and more than six-fold for under-five mortality compared to non-disabled mothers without HIV.

## Discussion

To our knowledge, this is the first epidemiological study examining the potential compounding effect of maternal disability status and maternal HIV status on child mortality in South

Africa. Our findings provide empirical evidence that maternal HIV and maternal disability have a compounded effect on child survival. We demonstrate quantitatively that in addition to maternal HIV- and maternal disability-based disparities in child mortality, children born to disabled mothers with HIV experience added child mortality disparities. The adjusted risk ratios of premature mortality among children born to disabled mothers with HIV compared to their peers born to non-disabled mothers without HIV were more than three times greater for neonatal and infant mortality and more than five times greater for under-five mortality. The effect was compounded, as the risk ratios for the combination of maternal disability and maternal HIV was more than the sum of each of the individual effects.

Our study also provides new epidemiological evidence on the robust association between maternal disability status and child mortality. The adjusted risk ratios for premature mortality among children born to disabled mothers compared to their peers born to non-disabled mothers were more than 70% greater for neonatal and under-five mortality and more than 65% for infant mortality. A growing body of research clearly demonstrates the higher risk for adverse maternal and birth outcomes experienced by disabled women worldwide. Consistent evidence shows that the adverse maternal and birth outcomes experienced by women with disabilities are not a result of their disabilities, but rather a combination of socially determined factors. For example, women with disabilities are at higher risk of multidimensional poverty, including lower rates of literacy, inaccessible health education and health care services, and lower employment rates, and other forms of social exclusion [6, 7, 17]. Further, disabled women also face stigma and discrimination both within their communities [18, 19] and too often within health care services [20–22]. These social vulnerabilities of disabled women, including gender-based violence and lack of access to prenatal care, play a significant role in premature child mortality risk.

While we found higher rates of child mortality among children with HIV-positive mothers, the adjusted risk ratios for child mortality indicators among children born to HIV-positive mothers were not statistically significant. A study published in 2005 [12] found that children had an excess of 2.9 (95% CI: 2.3–3.6) times higher risk for child mortality using pooled data from three longitudinal community-based studies from Uganda, Tanzania, and Malawi. The primary pathways for the excess risks of child mortality were the direct transmission of the virus to newborns by the infected mothers and indirectly through child mortality associated with maternal death [11, 12]. We posit two reasons for the lack of association between child mortality and maternal HIV in our study. First, it has been more than a decade since the earlier study was published. During this time, while the government of South Africa have been investing modestly in the prevention of mother-to-child transmission (PMTCT) and HIV and antiretroviral therapy (ART) treatment programs, there has been a great deal of NGO/UN-based efforts by both national and international groups throughout South Africa which may have also affected the rates [23–26]. Part of the improvements in child mortality among South African HIV-positive women can also be attributed to initiatives integrating the antiretroviral treatment (ART) services into the maternal and child health (MCH) services platform during the postnatal period. These initiatives have proved to be a simple and effective intervention for improving maternal and child outcomes in the context of HIV [24, 25, 27]. HIV transmission rates from mothers to children declined from 25–30% prior to 2001 to an estimated 1.4% in 2015 [28]. The use of longitudinal study and the inclusion of children whose mothers died is likely a second reason for the differential findings. Our study used cross-sectional data from SADHS based on the interviews only from surviving mothers. Hence, we cannot observe the indirect effect of maternal HIV on child mortality through maternal deaths. Future studies need to use longitudinal data to examine whether the non-significant differences in child mortality among HIV-positive attributable to the investment in the PMTCT and HIV treatment

programs by national governments and donor agencies in sub-Saharan Africa. Additional studies are needed to examine the context, etiology, and underlying mechanisms of the combined effect of maternal disability and maternal HIV on child mortality.

## Policy implications

South African government, donors, and NGOs may use findings from this study to measure progress in ongoing policy and program implementation efforts. Although South Africa has an impressive Constitution, some legislation and comprehensive disability policies [29–31], these are typically far behind in implementation. It also adopted and ratified the UN Convention on the Rights of Persons with Disabilities. South Africa has a disability grant program that pays an average equivalent of US $ 29 per month in 2021 for severely disabled persons and a framework for personal income tax deductions for costs pertaining to disabilities. Nevertheless, disability is common, poverty and inequality are widespread, and South Africa is far behind in implementing its own progressive policies and targets for persons with disabilities. This study findings provide empirical evidence that South African women with disabilities are sexually active and have children at rates comparable to those of their non-disabled peers. The inclusion of disability in the national strategic plans and a more disability-inclusive HIV service ecosystem in South Africa does not yet equally benefit women with disabilities, especially women with disabilities with HIV infection. Huge backlogs exist for assistive devices, employment targets for disabled persons are far from being achieved, and the Human Rights Commission, Public Protector and Department of Social Development have reported poor implementation [30]. Stigma against HIV-infected persons and persons with disabilities are substantial problems. This study reports startling differentials in key child health outcome indicators for disabled women and women with HIV infection. It highlights the need for better implementation of a range of policies, including those pertaining to access to the integrated health sector and maternity services.

A multipronged approach, supported through adequate government financing and donor-driven sector-wide approach in addition to the traditional project approach [32] needs to be used to improve access to maternity care for disabled women, women with HIV infection, and that most vulnerable group, disabled women with HIV infection. A systematic collection of epidemiological data segregated by all types of disabilities (e.g., individuals who are blind, deaf, those who have cognitive or physical disabilities, etc.) within the Government-wide monitoring and evaluation system [33] is critical to measure progress towards the national targets for disabled persons, inform policies, and implement inclusive programming. Outreach programs on sexual and reproductive health and other behavioral health communication topics should be adapted to meet women's needs with all types of disabilities. Healthcare systems at all levels, and especially family planning, prenatal, and postpartum care services should be accessible to all women with all types of disabilities. Given the longstanding societal beliefs and stigmas against women with disabilities and women with HIV infection in South Africa and the sub-Saharan context [5, 34], it is crucial to work with clinicians, obstetricians, midwives, nurses, and community health workers to help them recognize their explicit or implicit biases towards persons with disabilities and to develop tools and trainings to deal with them. South Africa, with its long-established network of Disabled Peoples Organizations, academic centers, and networks of disability scholars, its strong national disability legislation and social service disability programs, has a number of resources that can help provide such training and guidance for those working in the health sector. In addition to clinical and awareness-building approaches, programs to empower women with disabilities, especially those with both disabilities and HIV infection, become economically independent through primary and secondary

education, skills training, and preferential placement in the labor force in their approach in addressing the problem. Finally, systematic efforts are needed at levels to address the potential underlying mechanism for the association between maternal disability, maternal HIV, and child mortality, including the risk of gender-based violence and sexual abuse, poverty, lack of education and resources, social exclusion, stigma, and discrimination [34, 35].

## Limitations

There are several limitations to this study that are worth noting. First, the SADHS does not include questions on the severity, duration, onset, and cause of maternal disability—all of which may limit the sensitivity and accuracy of the data presented. Second, the data were self-reported and subject to potential recall and social desirability bias, likely leading to misclassification. Third, because this is a cross-sectional study, a causal relationship could not be determined. Future longitudinal research is needed to establish causal relationships. Finally, because only 52% of women and men aged 15 and older who were eligible for HIV testing were both interviewed and tested [14], the generalizability of the prevalence estimates is therefore unclear, and these results should be interpreted with caution.

## Conclusions

This study investigated the compounded risk of child mortality based on maternal HIV and maternal disability among children in South Africa. Our findings of compounded disparities among children born to women with disabilities with HIV infection suggest that these children may be at an exceptionally high risk of experiencing child mortality. Given the deleterious compounded disparities in child mortality, addressing the needs of women with disabilities, especially disabled women with HIV infection, and understanding the pathways and mechanisms contributing to these disparities is crucial. The findings are highly relevant to policy-makers, donor agencies, and non-governmental organizations working across various sectors to prevent child mortality and address the needs and rights of disabled women, women with HIV infection, or disabled women with HIV infection. Future research is needed to understand the interactions between maternal HIV infection and maternal disability and the underlying mechanisms through which these markers influence premature child mortality risk.

## Acknowledgments

The authors thank Clare L. Hurley of Brandeis University for editorial assistance.

## Author Contributions

**Conceptualization:** Ilhom Akobirshoev, Hussaini Zandam, Nora Groce, Mark Blecher, Monika Mitra.

**Data curation:** Ilhom Akobirshoev.

**Formal analysis:** Ilhom Akobirshoev, Hussaini Zandam.

**Funding acquisition:** Allyala Nandakumar.

**Investigation:** Ilhom Akobirshoev, Hussaini Zandam.

**Methodology:** Ilhom Akobirshoev, Hussaini Zandam, Monika Mitra.

**Software:** Ilhom Akobirshoev.

**Supervision:** Ilhom Akobirshoev.

**Validation:** Hussaini Zandam.

**Writing – original draft:** Ilhom Akobirshoev.

**Writing – review & editing:** Ilhom Akobirshoev, Hussaini Zandam, Allyala Nandakumar, Nora Groce, Mark Blecher, Monika Mitra.

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
