## [Decision Letter · Decision Letter 0]

29 Mar 2021

PONE-D-21-07389

The Compounding Effect of Having HIV and a Disability on Child Mortality among Mothers in South Africa

PLOS ONE

Dear Dr. Ilhom

Thank you for submitting your manuscript to PLOS ONE. After careful consideration, we feel that it has merit but does not fully meet PLOS ONE’s publication criteria as it currently stands. Therefore, we invite you to submit a revised version of the manuscript that addresses the points raised during the review process.

We look forward to receiving your revised manuscript.

Kind regards,

Godfrey Nyangadzayi Musuka

Academic Editor

PLOS ONE

Journal Requirements:

"I have read the journal's policy and the authors of this manuscript have the following

competing interests: Drs. Akobirshoev, Zadam and Nandakumar report receiving a

grant from Cardno Emerging Markets, USA, Ltd., for the conduct of the study."

Additional Editor Comments (if provided):

1.The fact that you there are no authors from South Africa who have a good understanding of the issues at hand in that country, I find it very difficult to believe that your research is meaningful and policy relevant in the country from where the data was collected.

2. What are the policy relevant issues from your study from a south Africa government perspective.

3. What are the policy implications for the SADC region as a whole?

4. The research is just an academic exercise and has no meaning if these fundamental questions are not answered by the researchers.

Reviewers' comments:

Reviewer's Responses to Questions

**Comments to the Author**

1. Is the manuscript technically sound, and do the data support the conclusions?

Reviewer #1: Yes

2. Has the statistical analysis been performed appropriately and rigorously? 

Reviewer #1: Yes

3. Have the authors made all data underlying the findings in their manuscript fully available?

Reviewer #1: Yes

4. Is the manuscript presented in an intelligible fashion and written in standard English?

Reviewer #1: Yes

5. Review Comments to the Author

Reviewer #1: A scientifically sound and well-thought piece of academic looking at the effects of having HIV and a disability on child mortality among mothers in South Africa. The statistical methods, data interpretation and discussion are well-laid out in a clear way.

6. PLOS authors have the option to publish the peer review history of their article (what does this mean?). If published, this will include your full peer review and any attached files.

Reviewer #1: **Yes: **Grant Murewanhema

---

## [Author Response · Author response to Decision Letter 0]

19 Apr 2021

EDITOR COMMENTS:

COMMENT #1: The fact that you there are no authors from South Africa who have a good understanding of the issues at hand in that country, I find it very difficult to believe that your research is meaningful and policy relevant in the country from where the data was collected.

RESPONSE #1:

Thank you for your comment. We really appreciate and fully support the idea of “Nothing about us without us.” We have invited Dr. Mark Blecher, Chief Director of the Health and Social Development at the National Treasury of South Africa to review our paper and contribute by highlighting the policy relevant issues from South Africa government perspective. Dr. Mark Blecher contribution was significant and we have added him as a co-author on this paper.

COMMENT #2: What are the policy relevant issues from your study from a south Africa government perspective.

RESPONSE #2:

Thank you for your comment. With the help from Dr. Mark Blecher we have addressed this in the text of the manuscript. We have now added a separate sub section “Policy implication” to the Discussion section and we suggest that the “South African government, donors, and NGOs may use findings from this study to measure progress in ongoing policy and program implementation efforts” in addressing the needs of women with disabilities, especially disabled women with HIV infection. 

COMMENT #3: What are the policy implications for the SADC region as a whole? The research is just an academic exercise and has no meaning if these fundamental questions are not answered by the researchers.

RESPONSE #3:

Thank you again for your comment. We have now addressed this in the Policy implications section. For example, we have stated that, “This study’s findings provide empirical evidence that South African women with disabilities are sexually active and have children at rates comparable to those of their non-disabled peers. The inclusion of disability in the national strategic plans and a more disability-inclusive HIV service ecosystem in South Africa does not yet equally benefit women with disabilities, especially women with disabilities with HIV infection.”

REVIEWER # 1 (DR. GRANT MUREWANHEMA):

COMMENT #1: A scientifically sound and well-thought piece of academic looking at the effects of having HIV and a disability on child mortality among mothers in South Africa. The statistical methods, data interpretation and discussion are well-laid out in a clear way.

RESPONSE #1: Thank you for taking your time to review our manuscript.

---

## [Editor Report · Decision Letter 1]

22 Apr 2021

The Compounding Effect of Having HIV and a Disability on Child Mortality among Mothers in South Africa

PONE-D-21-07389R1

Dear Dr. Ilhom

We’re pleased to inform you that your manuscript has been judged scientifically suitable for publication and will be formally accepted for publication once it meets all outstanding technical requirements.

Kind regards,

Godfrey Nyangadzayi Musuka

Academic Editor

PLOS ONE

---

## [Editor Report · Acceptance letter]

27 Apr 2021

PONE-D-21-07389R1 

The Compounding Effect of Having HIV and a Disability on Child Mortality among Mothers in South Africa 

Dear Dr. Akobirshoev:

I'm pleased to inform you that your manuscript has been deemed suitable for publication in PLOS ONE. Congratulations! Your manuscript is now with our production department. 

Kind regards, 

on behalf of

Dr. Godfrey Nyangadzayi Musuka 

Academic Editor

PLOS ONE